# Rearing condition influences gene expression in postlarval American lobster (*Homarus americanus*)

**Aubrey Jane[1], Douglas B. Rasher[2], Jesica Waller[3], Eric Annis[4], Markus Frederich[1]\***

1 University of New England, School of Marine and Environmental Programs, Biddeford, Maine, United States of America, 2 Bigelow Laboratory for Ocean Sciences, East Boothbay, Maine, United States of America, 3 Maine Department of Marine Resources, West Boothbay, Maine, United States of America, 4 Hood College, Frederick, Maryland, United States of America

\* mfrederich@une.edu

**Data Availability Statement:** Raw data have been deposited in GenBank Sequence Read Archive,

## Abstract

The American lobster (*Homarus americanus*) is an economically important species in the western Atlantic and its climate-driven range shift northward along the east coast of the United States is well documented. The thermal tolerance of lab-reared postlarvae of this species has been extensively investigated to better understand settlement and recruitment dynamics. However, there have been few studies focused on wild-caught postlarvae, and even fewer that have analyzed lab-rearing conditions in context of diet. In this study, we investigated gene transcriptional changes in postlarvae caught in the wild, as well as postlarvae reared in the laboratory on a brine shrimp diet or a wild-sourced zooplankton diet. We found between wild-caught and brine shrimp-reared larvae 3,682 differentially expressed genes, and between wild and zooplankton-reared postlarvae 3,939 differentially expressed genes. Between the two lab-reared groups fed different diets 2,603 genes were differentially expressed. We also exposed individuals in all rearing groups to chronic temperature treatments of 8°C and 26°C and found that both temperature extremes elicit 68–95% fewer transcriptional changes in wild postlarvae compared to either lab-reared group. In wild postlarvae, we identified differential expression of transcripts within the FoxO signaling pathway, a signaling pathway with a central role in cellular physiology, as potential molecular markers for cold tolerance in the American lobster. These findings contextualize the current literature on lab-reared postlarvae as containing conservative estimates for *in situ* organisms and can be used to inform population distribution modeling efforts. They also provide evidence for the need to adjust lab-rearing techniques or source wild larval crustaceans to augment studies of larval biology.

## Introduction

The American lobster (*Homarus americanus*) is an economically important species [1] with a well-documented climate-driven northward range shift along the coast of the eastern United

SRA (https://www.ncbi.nlm.nih.gov/sra), accession number PRJNA1087720.

**Funding:** This work was funded by National Science Foundation grants OCE-1947639, OCE-1948108, and OCE-1948146 to Markus Frederich, Eric Annis, and Doug Rascher, respectively. The funding organization did not play any role in the study design, data collection and analysis, decision to publish, or preparation of the manuscript.

**Competing interests:** The authors have declared that no competing interests exist.

States [2, 3]. The settlement and recruitment success of early life stages subsequently dictates population distribution [4–7]. After hatching, lobsters go through three planktonic larval stages with varying zonation in the water column [8], and a planktonic postlarval stage, before settling as stage V juveniles at the benthos. Stage IV postlarvae are estimated to spend up to 65% of their time in the upper 0.5 meters of the water column before settling to molt to stage V [9], and their distribution within rapidly warming surface waters leaves them more vulnerable to climate change than their deeper-dwelling (stages II and III) and benthic (stage V) counterparts [10, 11].

In fact, a disconnect between the abundance of egg-bearing females and the abundance of juveniles has put pre-settlement life stages, and specifically stage IV postlarvae, at the forefront of research on early development of this species [9, 12–17]. Recent recruitment failure has been attributed to sea surface temperature anomalies in the Gulf of Maine [18]. Other field-based studies have outlined how thermal tolerance, altered molting times, and food availability–especially at the extreme thermal ranges of the species' distribution–together influence *in situ* population dynamics [19]. In particular, the climate-driven decrease in copepod prey species *Calanus finmarchicus* has been investigated as a driving force behind recruitment failure [20].

Lab-based studies have the benefit of isolating individual factors, like temperature, to determine their influence on larval fitness and success. In such studies, it has been shown that increased seawater temperatures in early life are linked to increased mortality and decreased development time in crustaceans [21–24]. Further, the first signs of sub-lethal stress in lab-reared stage IV postlarvae occur from acute exposures to 26–33.8°C [25] and larval performance and growth are reduced after chronic exposure to temperatures of 23°C and above [21]. More contemporary lab-based studies have corroborated these findings, outlining that development time decreases with increasing temperature from 10°C to 22°C [26].

RNA-Seq is a useful and recently popularized tool for probing changes in gene expression in larval crustaceans, especially in the context of thermal tolerance and climate change [27]. Using this approach in a separate study, we have shown that the regulation of transcripts related to heat and UV tolerance is altered at different life stages in lab-reared American lobster. Stage IV postlarvae exhibit elevated expression of DnaJ homologs, which are notable for their involvement in the cellular response to heat and UV stress [28]. The ability of lab-reared postlarvae to reallocate energy to important physiological functions like preventing and repairing cellular damage is compromised under projected future climate warming scenarios [15, 17]. Additionally, projected warming scenarios, in conjunction with elevated $CO_2$, alter the regulation of genes related to shell-building processes and immune function in lab-reared postlarvae [16].

Historically, lab-based studies on lobster larvae have utilized larvae hatched in the lab and reared on excess, freshly hatched brine shrimp (*Artemia spp.*). These larvae are also typically reared at constant temperatures, usually between 15°C and 20°C, with 18°C yielding optimal growth rates for American lobster [29]. It is widely accepted that using lab-reared organisms reduces variability among individuals. Though this practice is commonplace, there is a growing body of evidence in crustaceans and other animals, that lab-reared individuals of many marine species may not be fully representative of their wild counterparts. In lab-reared fish, a diet of *Artemia spp.* yields nutritional deficiencies of essential fatty acids EPA/DHA and vitamin C [30]. Lab-reared fish are less efficient at catching prey and avoiding predators than their wild counterparts [31, 32]. In *Homarus americanus*, similar results have been found, with swimming speeds of 18 cm/s observed in wild postlarvae compared to swimming speeds of 10 cm/s observed in lab-reared postlarvae [13]. Brine shrimp-fed postlarvae are also known to be

deficient in polyunsaturated fatty acyl chains, and one study suggests that copepods may be worth investigating as an alternative feed for lobster hatcheries [14].

To identify cellular processes driving the observed differences between lab-reared and wild-caught *Homarus americanus* postlarvae, we investigated wild-caught postlarvae, lab-reared postlarvae fed the traditional diet (freshly hatched brine shrimp), and lab-reared postlarvae fed freshly caught zooplankton. We applied transcriptomics to identify the reorganization of cellular pathways as a result of rearing condition. Additionally, we subjected individuals from each of these conditions to a lower (8°C) and upper (26°C) thermal exposure to see how each rearing group's molecular thermal response varied. To the best of our knowledge, our study is the first to investigate changes in gene expression of *H. americanus* postlarvae in response to different rearing conditions.

## Methods

### Animal husbandry

Eight ovigerous female lobsters were sourced from lobster management zone E in the Gulf of Maine in the summer of 2022 through the State of Maine Department of Marine Resources ventless trap survey. The ovigerous lobsters were held under State of Maine Department of Marine Resources Special License #2022-19-04 at the Bigelow Laboratory for Ocean Sciences in East Boothbay, Maine, in a common flow-through tank fed by the Damariscotta River estuary (average tank temperature 14.4°C; salinity 30–32 ppt). Hatchlings were fed within 1 to 10 hours of hatch and isolated within 24 hours in individual 400 mL glass jars filled with 0.45 μm filtered sea water. Jarred individuals were acclimated to the documented optimal rearing temperature of 18°C [29], and this temperature served as a control in our experiments. Approximately 90% of the water in each jar was changed three times per week. Water was not aerated, but regular water changes yielded average dissolved oxygen levels of 7.2 mg/L.

### Rearing condition experiments

A subset (n = 5) of lab-reared larvae were fed freshly hatched *Artemia* sp. ad libitum, the traditional diet fed to larval lobsters in a research setting. Another subset (n = 5) of lab-reared larvae were fed a diet of freshly caught zooplankton ad libitum, representative of their natural diet. On average, postlarvae fed zooplankton were 13% larger and 435% heavier than the postlarvae fed brine shrimp (weight: wild-caught 26.3 +/- 9.20 mg, brine shrimp fed 4.9 +/- 0.90 mg, zooplankton fed 5.6 +/- 0.40 mg; carapace length wild-caught 6.16 +/- 0.53 mm, brine shrimp fed 3.94 +/- 0.54 mm, zooplankton fed 3.87 +/- 0.19 mm). Zooplankton was caught in vertical tows within 48 hours of feeding in the same area the mothers were sourced from. These tows, on average, were comprised of 62% copepods, including *Calanus finmarchicus*, a potential prey species whose abundance is significantly correlated to successful lobster settlement [20]. Additionally, fish eggs and other larval crustaceans were captured. Most of the contents of our zooplankton tows have been documented as prey items in gut content analyses of larval lobsters [12]. Lab-reared lobsters from both feeding conditions were sacrificed 48 hours after molt to stage IV. Wild stage IV lobsters, identified based on morphology (n = 5) were also captured in lobster management zone E using a neuston net with 1 mm mesh [1] and held in the lab at ambient temperature (temperature of the water in the flow-through tank fed by the Damariscotta River estuary; average temperature 14.4°C (minimum 11.0°C, maximum 19.6°C) for 48 hours before being preserved for analysis. Wild postlarvae were, on average, 13% larger than their brine shrimp-reared counterparts.

## Temperature treatments

A subset of lobsters from each rearing condition (n = 5 per group) was subjected to a low-temperature treatment of 8°C and another subset at a high-temperature treatment of 26°C for 48 hours, except for zooplankton-fed postlarvae, which were only analyzed for cold tolerance. These temperature treatments resemble upper and lower extremes that these larvae experience in their habitats. Lab-reared lobsters were acclimated to their respective temperature treatments within 24 hours after molt to stage IV and maintained at these temperatures for 48 hours until sampling. Wild-caught stage IV lobsters were immediately acclimated to their respective temperature treatments upon being brought back to the lab and maintained at their respective temperatures for 48 hours before sampling. Sampling time points were kept consistent so lobsters in rearing condition experiments could be used as controls.

## Sample preservation

Sampled lobsters (n = 5 per rearing condition, per temperature treatment) were rinsed with milli-Q water before preservation in DNA-RNA-free microcentrifuge tubes containing 600 μL of RNAlater solution (Ambion Inc.). Samples were held at 8°C for 12 hours to allow the preservative to permeate the tissue before being transferred to -80°C for storage until analysis. Samples were shipped on dry ice to Novogene (Sacramento, CA) for extraction, sequencing, and analysis, following their internal protocols.

## Library preparation

Library preparation and differential expression analysis were performed as described in [28]. In brief: Libraries were prepared using poly-A enrichment to select for mRNA following Novogene's internal quality control assays. Sequencing was done using the Illumina NovaSeq platform. Samples were sequenced as paired-end reads of 150 base pairs, and those reads in the fastq format were processed for downstream analyses or "cleaned" by removing reads containing adaptors, reads with >10% uncertain nucleotides, and reads with >50% low quality nucleotides, defined as Base Quality < 5. Reads were aligned to a reference sequence for American lobster [33] using HISAT2 (version 2.0.5) and assembled into transcripts using StringTie (version 1.3.3b). Raw counts of map reads were aggregated using featureCounts (version 1.5.0-p3). Raw data have been deposited in GenBank Sequence Read Archive, SRA, accession number PRJNA1087720.

## Differential expression analysis

Raw read counts were normalized to correct for sequencing depth before differential transcript expression was analyzed using DESeq2 software [34]. DESeq2 uses the negative binomial as the reference distribution and takes a geometric normalization approach. False discovery rate (FDR) was controlled by adjusting p-values for multiple testing with the Benjamini-Hochberg procedure, and the differential transcript screening threshold was $|\log2(FoldChange)| \geq 1$ and padj $\leq 0.05$. Gene Ontology (GO) and Kyoto Encyclopedia of Genes and Genomes (KEGG) pathway enrichment analysis was performed using clusterProfiler (version 3.8.1) [35, 36]. The KEGG pathway enrichment analysis was restricted solely to *Homarus americanus* (hame).

# Results

## Transcriptome quality and mapping

Samples showed consistent read quality and depth. Across treatments, 459,499,224 raw reads were generated, of which, an average of 97.4% were clean reads and 91.8% were able to be

Table 1. RNA-seq read quality parameters of *Homarus americanus* larvae in different rearing conditions (n = 5 per treatment group).

| Treatment group | Average raw reads | Clean reads (%) | Reads aligned to genome (%) |
|---|---|---|---|
| Brine shrimp-reared (18°C) | 56246590 | 97.0 | 92.6 |
| 8°C | 67756184 | 97.4 | 92.5 |
| 26°C | 57717047 | 97.5 | 92.5 |
| Wild caught (ambient) | 53874813 | 97.3 | 92.4 |
| 8°C | 62163368 | 97.5 | 92.6 |
| 26°C | 55856186 | 97.6 | 90.9 |
| Zooplankton-reared (18°C) | 45114401 | 97.4 | 91.1 |
| 8°C | 60770635 | 97.4 | 89.7 |
| Overall: | 459499224 | 97.4 | 91.8 |

aligned with the reference genome (Table 1). Mapping gene IDs to the GO and KEGG databases identified 2,921 unique GO terms and 133 unique KEGG pathways represented in our transcriptome.

## Differential transcript expression among treatment groups

In wild-caught postlarvae compared to brine shrimp-reared postlarvae, 1,406 transcripts were over-expressed, and 2,276 transcripts were under-expressed. In zooplankton-reared postlarvae relative to brine shrimp-reared postlarvae, 1,534 transcripts were over-expressed, and 1,069 transcripts were under-expressed. In zooplankton-reared relative to wild-caught, 2,795 transcripts were over-expressed, and 1,144 transcripts were under-expressed (Fig 1).

Wild-caught larvae exhibited 95% fewer differentially expressed transcripts in response to cold exposure compared to brine shrimp-reared postlarvae, and 86% fewer than zooplankton-reared postlarvae. Zooplankton-reared postlarvae exhibited 64% fewer differentially expressed transcripts in response to cold exposure than brine shrimp-reared postlarvae (Fig 2). Across the cold-treated rearing groups, 36 transcripts were commonly differentially expressed, including transcripts for DNA helicase, methyltransferase, and polymerase, among others. The full list of genes, their expression values, and results from statistical comparisons among groups can be found in S1 File. Variability from sample to sample was relatively consistent among groups, and hierarchical clustering of individual samples in relation to common differentially expressed genes yielded distinct groupings of the known experimental variables. The first tier of branching separated samples by temperature treatment. On the second tier, cold-treated postlarvae were further separated by rearing environment, with one cluster of wild postlarvae and another cluster of lab-reared postlarvae. The cluster of lab-reared postlarvae branched further into two separate groups defined by their different diets. The postlarvae that were not cold-treated were separated by diet first, with brine shrimp-reared postlarvae in a separate cluster from zooplankton-reared and wild postlarvae. Wild postlarvae all clustered together; one zooplankton-reared postlarvae grouped with the brine shrimp-reared postlarvae (Fig 3).

Wild postlarvae exhibited 68% fewer differentially expressed genes in response to heat exposure compared to brine shrimp-reared postlarvae (Fig 4). In total, 152 transcripts were commonly differentially expressed across rearing groups in response to heat exposure. The full list of genes, their expression values, and results from statistical comparisons among groups can be found in S2 File. Again, variability from sample to sample was relatively consistent among groups, and hierarchical clustering of individual samples yielded an initial branching

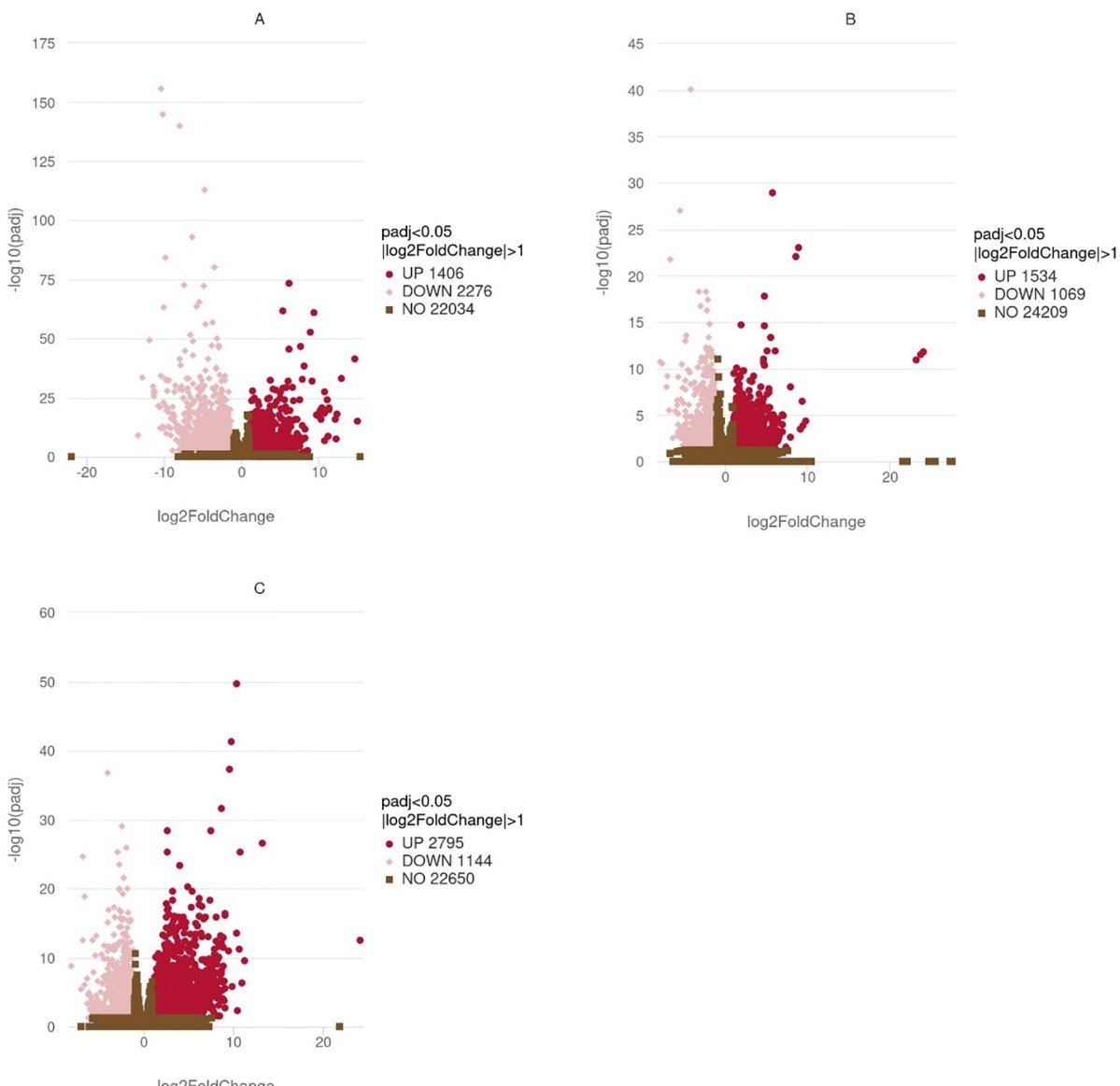

**Fig 1. Volcano plots show differential gene expression in *Homarus americanus* stage IV post larvae after different rearing conditions.** A: wild-caught postlarvae vs brine shrimp-reared postlarvae, B: zooplankton-reared postlarvae vs brine shrimp-reared postlarvae, C: zooplankton-reared postlarvae vs wild-caught postlarvae, all at rearing temperatures. The blue dashed line indicates the threshold for differential gene screening criteria. Red dots represent up-regulated genes and pink dots represent down-regulated genes. Axes are scaled differently in each plot for granularity.

off of samples within the different temperature treatments, and then further branching of samples within different rearing groups (Fig 5).

## GO term and KEGG pathway enrichment analysis

**Wild-caught vs. brine shrimp-reared postlarvae.** There were 28 GO terms and 8 KEGG pathways significantly enriched in the comparison between wild-caught and brine shrimp-reared postlarvae reared at ambient temperature, and these two groups were the least similar

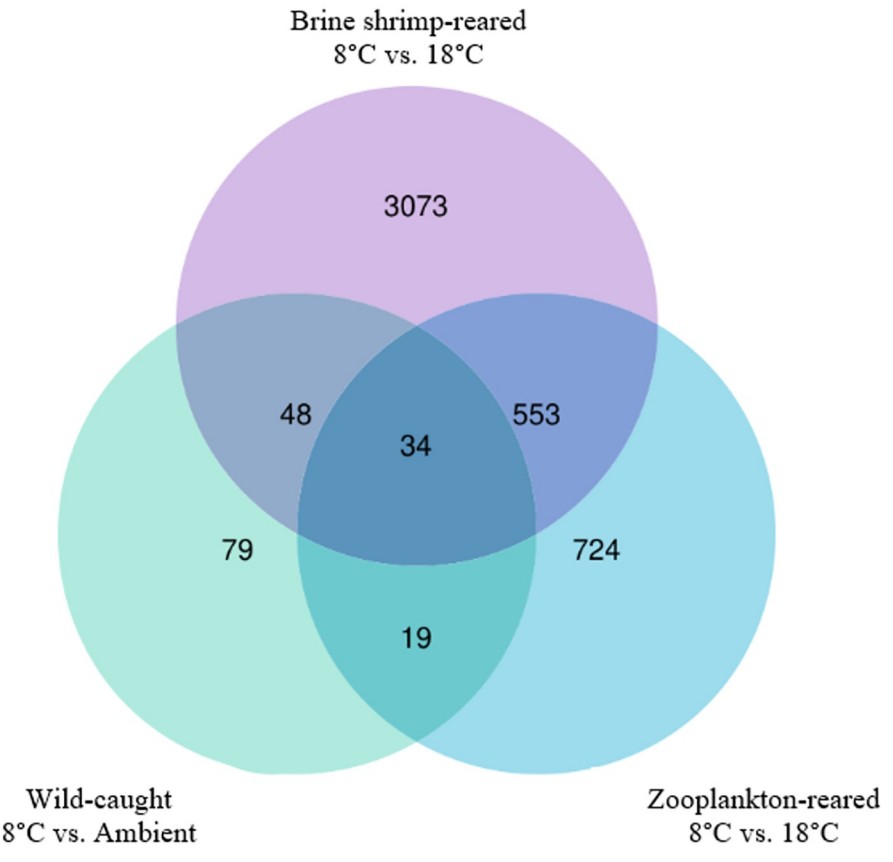

**Fig 2. Venn diagram of differentially expressed transcripts in *Homarus americanus* stage IV postlarvae in response to cold exposure.** 3,073 transcripts were uniquely differentially expressed in brine shrimp-reared postlarvae. 724 transcripts were uniquely differentially expressed in zooplankton-reared postlarvae. Only 79 transcripts were uniquely differentially expressed in wild-caught postlarvae. 32 transcripts were commonly expressed.

based on the enrichment analysis. Differentially expressed transcripts within the top four significantly enriched GO terms were largely under-expressed in wild-caught postlarvae; these terms, in order of statistical significance, were (1) structural constituent of cuticle, (2) extracellular region, (3) chitin binding, and (4) structural molecule activity. Differentially expressed transcripts in GO term (5) carbohydrate metabolic process were largely over-expressed in wild-caught postlarvae. The top five significantly enriched KEGG pathways included (1) mannose type O-glycan biosynthesis, (2) amino sugar and nucleotide sugar metabolism, (3) tyrosine metabolism, (4) glycosphingolipid biosynthesis—globo and isoglobo series, and (5) glycosaminoglycan degradation. Within all of these pathways, the majority of differentially expressed transcripts were over-expressed in wild-caught postlarvae (Tables 2 and 3).

**Zooplankton-reared vs. brine shrimp-reared postlarvae.** There were 31 GO terms and 6 KEGG pathways significantly enriched in the comparison between zooplankton-reared and brine shrimp-reared postlarvae reared at ambient temperature. The top five significantly enriched GO terms were (1) chitin binding, (2) extracellular region, (3) ligand-gated ion channel activity, and (4) ligand-gated channel activity, in which the majority of differentially expressed transcripts were under-expressed in zooplankton-reared postlarvae, and (5) carbohydrate metabolic process, in which the majority of differentially expressed transcripts were over-expressed in zooplankton-reared postlarvae. The top five most significantly enriched

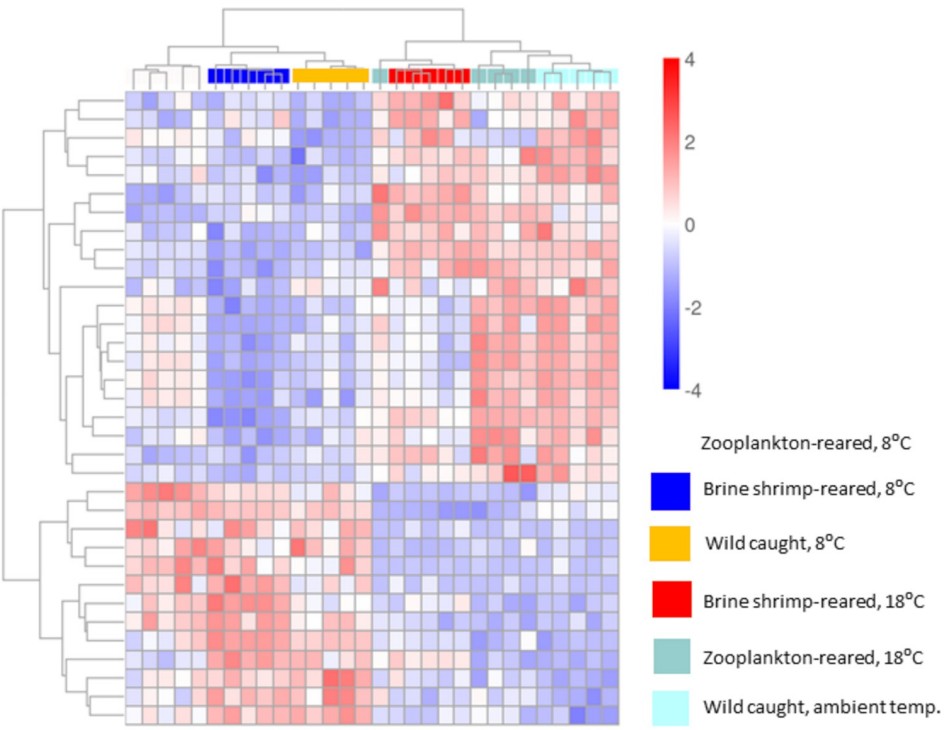

**Fig 3. Clustering heatmap shows the expression of commonly differentially expressed transcripts in *Homarus americanus* stage IV postlarvae in response to cold exposure.** At the top of the grid, boxes of varying colors represent samples within each experimental group, as follows (noted from left to right): cold-treated zooplankton-reared postlarvae in white; cold-treated brine shrimp-reared postlarvae in dark blue; cold-treated wild postlarvae in yellow; control brine shrimp-reared postlarvae in red; control zooplankton-reared postlarvae in green; control wild postlarvae in light blue. Mainstream hierarchical clustering was used to cluster the fpkm values of genes and homogenize each row (Z-score). The bar at right shows that, within the grid, a fourfold increase in expression compared to the control is represented by the darkest red, while a fourfold decrease in expression compared to the control is represented by the darkest blue. All samples within groups cluster together, except for one control zooplankton-reared postlarva which clusters with the control brine shrimp-reared postlarvae. For individual genes see S1 File.

KEGG pathways were (1) tyrosine metabolism, (2) amino sugar and nucleotide sugar metabolism, (3) lysosome, (4) glycine, serine and threonine metabolism, and (5) pentose and glucuronate interconversions, all of which contained differentially expressed transcripts which were largely over-expressed in zooplankton-reared postlarvae.

**Zooplankton-reared vs. wild-caught postlarvae.** There were 22 GO terms and one KEGG pathway significantly enriched in the comparison between zooplankton-reared and wild-caught postlarvae reared at ambient temperature, and these two groups were the most similar based on our enrichment analysis. The top five significantly enriched GO terms were (1) extracellular region, (2) endopeptidase activity, (3) signal transduction, (4) signaling, and (5) cell communication. Of these, endopeptidase activity was the only term containing largely under-expressed, rather than over-expressed, transcripts in zooplankton-reared postlarvae. The sole significantly enriched KEGG pathway, neuroactive ligand-receptor interaction, contained differentially expressed transcripts which were largely over-expressed in zooplankton-reared postlarvae compared to wild-caught postlarvae.

**Cold temperature exposure (8°C).** There were 6 GO terms and 9 KEGG pathways significantly enriched in the comparison between brine shrimp-reared postlarvae exposed to the cold, 8°C temperature treatment and those maintained at their rearing temperature (18°C).

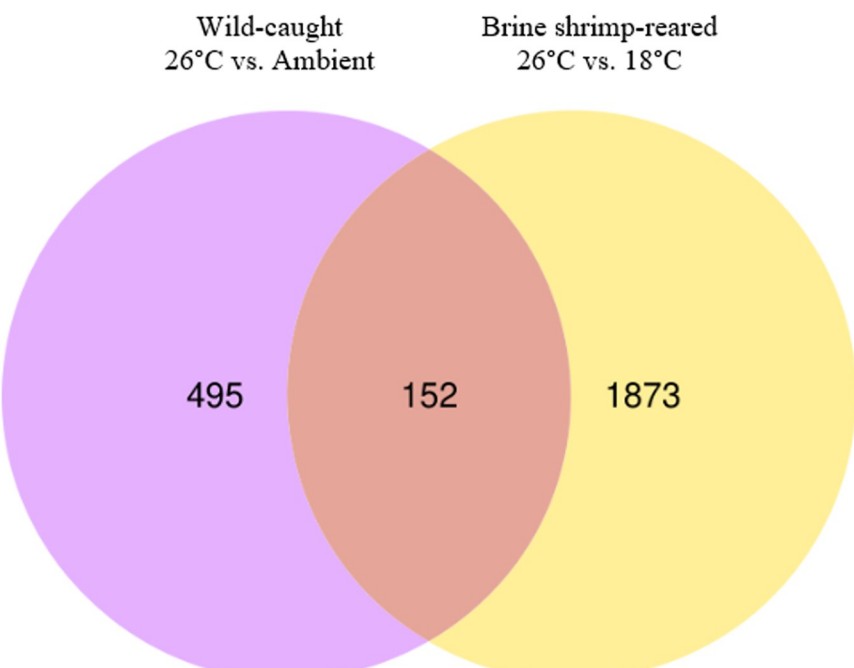

Wild-caught
26°C vs. Ambient

Brine shrimp-reared
26°C vs. 18°C

495    152    1873

**Fig 4. Venn diagram of differentially expressed transcripts in *Homarus americanus* stage IV postlarvae in response to heat exposure.** 495 transcripts were uniquely differentially expressed in wild-caught postlarvae. 1,873 transcripts were uniquely differentially expressed in brine shrimp-reared postlarvae. 152 transcripts were commonly expressed.

The top five significantly enriched GO terms were (1) structural constituent of cuticle, (2) extracellular region, (3) chitin binding, (4) G-protein coupled receptor activity, and (5) G-protein coupled receptor signaling pathway. The majority of differentially expressed transcripts within these terms were under-expressed in cold-treated (8°C) brine shrimp-reared postlarvae. The top five significantly enriched KEGG pathways were (1) DNA replication and (2) protein processing in endoplasmic reticulum, which contained transcripts that were largely under-expressed in response to cold exposure. Additionally, pathways for (3) purine metabolism, (4) pyrimidine metabolism, and (5) nucleotide metabolism all contained transcripts that were largely over-expressed in brine shrimp-reared postlarvae in response to cold exposure.

There were 35 GO terms and 4 KEGG pathways significantly enriched in the comparison between zooplankton-reared postlarvae exposed to the cold, 8°C temperature treatment and those maintained at their rearing temperature (18°C). The top five significantly enriched GO terms were (1) structural constituent of cuticle, (2) structural molecule activity, (3) extracellular region, and (4) chitin binding. Transcripts within these terms were largely over-expressed in the 8°C treatment compared to the control. Transcripts within (5) carbohydrate metabolic process were largely under-expressed in the 8°C treatment compared with the control. Exposure to 8°C in zooplankton-reared postlarvae also yielded the overarching under-expression of transcripts within 4 KEGG pathways; (1) glycolysis / gluconeogenesis, (2) DNA replication, (3) glycine, (4) serine and threonine metabolism, and (5) cysteine and methionine metabolism.

There was only one GO term and one KEGG pathway significantly enriched in the comparison between wild-caught postlarvae exposed to the cold, 8°C temperature treatment and those maintained at their rearing temperature (ambient, 11–19°C). The differentially expressed transcripts within the GO term, carbohydrate biosynthetic process were all over-

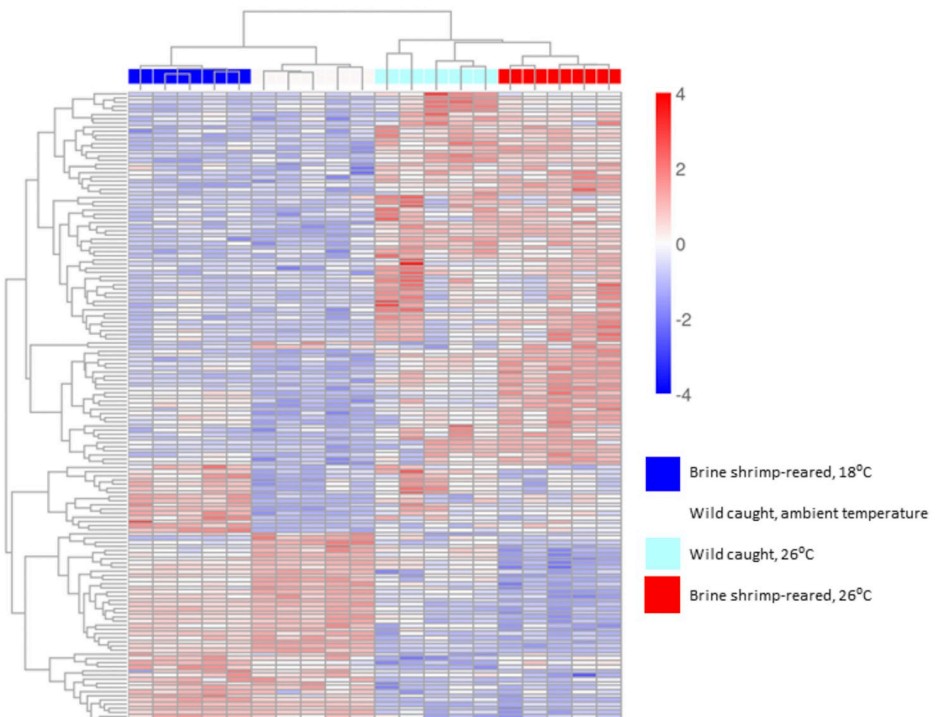

**Fig 5. Clustering heatmap shows the expression of commonly differentially expressed transcripts in *Homarus americanus* stage IV postlarvae in response to heat exposure.** At the top of the grid, boxes of varying colors represent samples within each experimental group, as follows (noted from left to right): control brine shrimp-reared postlarvae in dark blue; control wild postlarvae in white; heat-treated wild postlarvae in light blue; heat-treated brine shrimp-reared postlarvae in red. Mainstream hierarchical clustering was used to cluster the fpkm values of genes and homogenize each row (Z-score). The bar at right shows that, within the grid, a fourfold increase in expression compared to the control is represented by the darkest red, while a fourfold decrease in expression compared to the control is represented by the darkest blue. All samples within groups cluster together, except for one control zooplankton-reared postlarva which clusters with the control brine shrimp-reared postlarvae. For individual genes see S2 File.

expressed in response to cold exposure in wild postlarvae. In the KEGG pathway for FoxO signaling, two transcripts were under-expressed, and two transcripts were over-expressed in response to cold exposure in wild postlarvae. Those that were overexpressed were further analyzed for potential use as markers for cold tolerance; they were both transcripts for phosphoenolpyruvate carboxykinase, an enzyme involved in gluconeogenesis (for sequences and fpkm values see S3 and S4 Files). The differential expression of these two transcripts was reduced the further removed rearing conditions were from what is experienced in the wild (Fig 6).

**Warm temperature exposure (26°C).** There were 17 GO terms and 12 KEGG pathways significantly enriched in the comparison between brine shrimp-reared postlarvae exposed to the warm, 26°C temperature treatment and those maintained at their rearing temperature (18°C). The top five significantly enriched GO terms were (1) structural constituent of cuticle, (2) structural molecule activity, (3) extracellular region, (4) chitin binding, and (5) endoplasmic reticulum. Most transcripts within these terms were under-expressed in warm-treated (26°C) brine shrimp-reared postlarvae. The top five significantly enriched KEGG pathways in this comparison were (1) carbon metabolism, (2) glycolysis/gluconeogenesis, (3) starch and sucrose metabolism, and (4) amino sugar and nucleotide sugar metabolism, all of which contained transcripts which were largely under-expressed in response to heat exposure. Genes

**Table 2. Summary of GO term enrichment analysis of *Homarus americanus* postlarvae in different rearing conditions (n = 5 per treatment group).**

| | Comparison | Total # enriched GO terms | Category | GO term ID | Description | Gene Ratio | p-adjusted | # Upregulated Genes | # Downregulated Genes |
|---|---|---|---|---|---|---|---|---|---|
| Rearing condition | Zooplankton- vs. brine shrimp-reared | 31 | MF | GO:0008061 | chitin binding | 53/826 | 1.55E-09 | 24 | 29 |
| | | | CC | GO:0005576 | extracellular region | 61/306 | 9.36E-08 | 27 | 34 |
| | | | BP | GO:0005975 | carbohydrate metabolic process | 50/466 | 1.49E-07 | 41 | 9 |
| | | | MF | GO:0015276 | ligand-gated ion channel activity | 78/826 | 2.79E-05 | 17 | 61 |
| | | | MF | GO:0022834 | ligand-gated channel activity | 78/826 | 2.79E-05 | 17 | 61 |
| | Wild-caught vs. brine shrimp-reared | 28 | MF | GO:0042302 | structural constituent of cuticle | 132/1164 | 1.85E-28 | 15 | 117 |
| | | | CC | GO:0005576 | extracellular region | 102/362 | 2.76E-28 | 28 | 74 |
| | | | MF | GO:0008061 | chitin binding | 83/1164 | 4.30E-21 | 21 | 62 |
| | | | MF | GO:0005198 | structural molecule activity | 136/1164 | 2.99E-16 | 16 | 120 |
| | | | BP | GO:0005975 | carbohydrate metabolic process | 60/613 | 9.20E-08 | 45 | 15 |
| | Zooplankton-reared vs. wild-caught | 22 | CC | GO:0005576 | extracellular region | 90/616 | 6.78E-05 | 67 | 23 |
| | | | MF | GO:0004175 | endopeptidase activity | 81/1717 | 1.42E-03 | 30 | 51 |
| | | | BP | GO:0007165 | signal transduction | 146/950 | 4.37E-03 | 99 | 47 |
| | | | BP | GO:0023052 | signaling | 149/950 | 4.37E-03 | 100 | 49 |
| | | | BP | GO:0007154 | cell communication | 149/950 | 4.37E-03 | 101 | 48 |
| Temperature treatment | 8°C vs. 18°C (brine shrimp-reared) | 6 | MF | GO:0042302 | structural constituent of cuticle | 132/1256 | 2.02E-14 | 114 | 18 |
| | | | CC | GO:0005576 | extracellular region | 68/409 | 1.86E-04 | 54 | 14 |
| | | | MF | GO:0008061 | chitin binding | 56/1256 | 7.59E-04 | 46 | 10 |
| | | | MF | GO:0004930 | G-protein coupled receptor activity | 63/1256 | 6.66E-03 | 56 | 7 |
| | | | BP | GO:0007186 | G-protein coupled receptor signaling pathway | 67/674 | 8.02E-03 | 60 | 7 |
| | 26°C vs. 18°C (brine shrimp-reared) | 17 | MF | GO:0042302 | structural constituent of cuticle | 78/638 | 1.08E-14 | 2 | 76 |
| | | | MF | GO:0005198 | structural molecule activity | 81/638 | 3.22E-09 | 2 | 79 |
| | | | CC | GO:0005576 | extracellular region | 43/175 | 8.29E-08 | 12 | 31 |
| | | | MF | GO:0008061 | chitin binding | 34/638 | 3.54E-04 | 8 | 26 |
| | | | CC | GO:0005783 | endoplasmic reticulum | 11/175 | 2.70E-03 | 1 | 10 |
| | 8°C vs. ambient (wild-caught) | 1 | BP | GO:0016051 | carbohydrate biosynthetic process | 3/323 | 5.30E-03 | 3 | 0 |
| | 26°C vs. ambient (wild-caught) | 0 | - | - | - | - | - | - | - |
| | 8°C vs. 18°C (zooplankton-reared) | 35 | MF | GO:0042302 | structural constituent of cuticle | 87/467 | 1.58E-27 | 83 | 4 |
| | | | MF | GO:0005198 | structural molecule activity | 87/467 | 4.77E-19 | 83 | 4 |
| | | | CC | GO:0005576 | extracellular region | 49/130 | 4.56E-17 | 38 | 11 |
| | | | MF | GO:0008061 | chitin binding | 42/467 | 6.58E-12 | 37 | 5 |
| | | | BP | GO:0005975 | carbohydrate metabolic process | 27/229 | 5.13E-04 | 10 | 17 |

**Table 3. Summary of KEGG pathway enrichment analysis of *Homarus americanus* postlarvae in different rearing conditions (n = 5 per treatment group).**

| | Comparison | Total # enriched KEGG pathways | KEGG pathway ID | Description | Gene Ratio | p-adjusted | # Upregulated Genes | # Downregulated Genes |
|---|---|---|---|---|---|---|---|---|
| Rearing condition | Zooplankton- vs. brine shrimp-reared | 6 | hame00350 | Tyrosine metabolism | 17/306 | 5.79E-08 | 16 | 1 |
| | | | hame00520 | Amino sugar and nucleotide sugar metabolism | 22/306 | 2.45E-05 | 17 | 5 |
| | | | hame04142 | Lysosome | 31/306 | 2.49E-03 | 26 | 5 |
| | | | hame00260 | Glycine, serine and threonine metabolism | 11/306 | 1.28E-02 | 8 | 3 |
| | | | hame00040 | Pentose and glucuronate interconversions | 9/306 | 3.11E-02 | 8 | 1 |
| | Wild-caught vs. brine shrimp-reared | 8 | hame00515 | Mannose type O-glycan biosynthesis | 20/356 | 3.55E-07 | 18 | 2 |
| | | | hame00520 | Amino sugar and nucleotide sugar metabolism | 24/356 | 2.18E-05 | 16 | 8 |
| | | | hame00350 | Tyrosine metabolism | 12/356 | 3.48E-03 | 11 | 1 |
| | | | hame00603 | Glycosphingolipid biosynthesis—globo and isoglobo series | 10/356 | 1.14E-02 | 8 | 2 |
| | | | hame00531 | Glycosaminoglycan degradation | 9/356 | 1.60E-02 | 7 | 2 |
| | Zooplankton-reared vs. wild-caught | 1 | hame04080 | Neuroactive ligand-receptor interaction | 58/643 | 4.72E-02 | 35 | 23 |
| Temperature treatment | 8°C vs. 18°C (brine shrimp-reared) | 9 | hame03030 | DNA replication | 16/459 | 3.55E-04 | 0 | 16 |
| | | | hame04141 | Protein processing in endoplasmic reticulum | 37/459 | 3.55E-04 | 2 | 35 |
| | | | hame00230 | Purine metabolism | 28/459 | 1.78E-03 | 17 | 11 |
| | | | hame00240 | Pyrimidine metabolism | 15/459 | 1.84E-02 | 8 | 7 |
| | | | hame01232 | Nucleotide metabolism | 19/459 | 2.22E-02 | 12 | 7 |
| | 26°C vs. 18°C (brine shrimp-reared) | 12 | hame01200 | Carbon metabolism | 23/291 | 7.30E-04 | 5 | 18 |
| | | | hame00010 | Glycolysis / Gluconeogenesis | 15/291 | 7.30E-04 | 5 | 10 |
| | | | hame00531 | Glycosaminoglycan degradation | 9/291 | 5.68E-03 | 5 | 4 |
| | | | hame00500 | Starch and sucrose metabolism | 11/291 | 8.66E-03 | 5 | 6 |
| | | | hame00520 | Amino sugar and nucleotide sugar metabolism | 16/291 | 1.10E-02 | 5 | 11 |
| | 8°C vs. ambient (wild-caught) | 1 | hame04068 | FoxO signaling pathway | 4/223 | 1.16E-02 | 2 | 2 |
| | 26°C vs. ambient (wild-caught) | 0 | - | - | - | - | - | - |
| | 8°C vs. 18°C (zooplankton-reared) | 4 | hame00010 | Glycolysis / Gluconeogenesis | 10/161 | 1.07E-02 | 2 | 8 |
| | | | hame03030 | DNA replication | 8/161 | 1.09E-02 | 0 | 8 |
| | | | hame00260 | Glycine, serine and threonine metabolism | 8/161 | 1.35E-02 | 3 | 5 |
| | | | hame00270 | Cysteine and methionine metabolism | 7/161 | 3.83E-02 | 2 | 5 |

within the pathway (5) glycosaminoglycan degradation were largely over-expressed as a result of heat exposure. In stark contrast to the brine shrimp-reared postlarvae, exposure of wild-caught postlarvae to 26°C yielded no significantly enriched GO terms or KEGG pathways when compared with the control (ambient, 11–19°C).

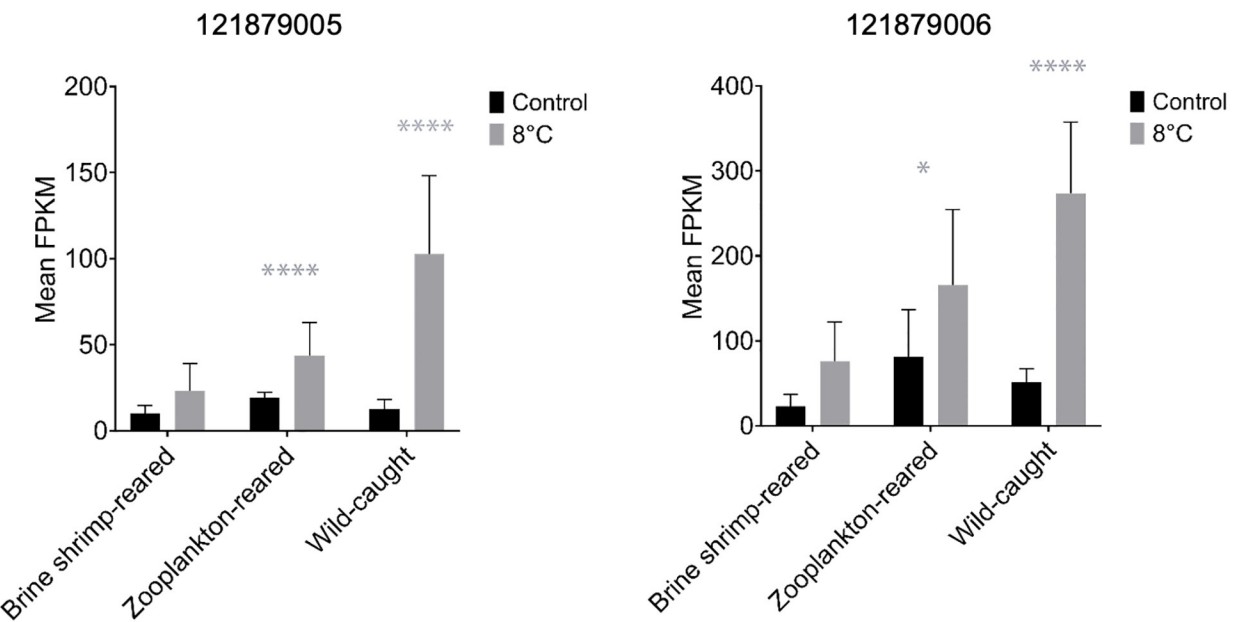

**Fig 6. Expression of two separate transcripts for a gene in *Homarus americanus* coding for phosphoenolpyruvate carboxykinase- an enzyme involved in gluconeogenesis.** 121879005 on the left, and 121879006 on the right. Mean normalized read counts are on the y-axes, and rearing conditions are on the x-axes; control temperature treatments are indicated by black bars, and cold temperature treatments are indicated by grey bars. Error bars represent standard deviation values. One star indicates a p-value of <0.05. Four stars indicate a p-value of <0.0001 (two-way ANOVA multiple comparisons, α = 0.05). The over-expression of both transcripts in cold treatments was reduced in rearing conditions and further removed from natural conditions.

## Discussion

We demonstrated that both diet (traditional lab-rearing diet of brine shrimp vs. "natural" diet of zooplankton) and environment (lab setting vs. ocean environment) influence the baseline whole-body gene expression of *Homarus americanus* at 18˚C in lab-reared postlarvae and at ambient in wild-caught, as well as transcriptional responses to temperature extremes. One cellular event that may drive differences in brine shrimp-reared postlarvae compared to wild-caught postlarvae is the under-expression of transcripts related to carbohydrate metabolic activity. This pattern likely reflects the brine shrimp-reared postlarvae's reduced energy availability and/or a reduction in energetic demands due to decreased predation, room to swim, or need to chase after scarce food. Transcripts related to amino sugar and nucleotide sugar metabolism, as well as tyrosine metabolism, were also under-expressed in brine shrimp-reared postlarvae, again highlighting their reduced metabolic demands when compared to wild-caught. Therefore, we interpret these findings as lab-reared postlarvae being metabolically adapted to the less demanding conditions of their upbringing, leading to greater responses to stress, as outlined below. Transcripts related to glycosaminoglycan degradation as well as the biosynthesis of mannose type O-glycans and glycosphingolipids, notable for their importance in cell-to-cell communication [37], were largely under-expressed in brine shrimp-reared postlarvae. Additionally, there were several transcripts related to shell-building processes (i.e., "structural constituents of the cuticle" and "chitin-binding") that were over-expressed in brine shrimp-reared postlarvae compared to wild-caught ones. This could be explained by the varied time points that wild lobsters were sampled (i.e., wild lobsters may have been at a point in their molt cycle in which shell-building processes were not upregulated compared with the lab-reared lobsters sampled 48 hours post-molt). While we chose to sample lobsters at the

48-hour time point in the interest of consistency for temperature exposure experiments, future studies may benefit from employing one of two sampling methods; randomizing the sampling of lab-reared lobsters to mimic fishing for wild postlarvae or staging lobsters and sampling groups only from the same molt stage, which has been done in other crustacean species [38, 39]. Additionally, future studies may benefit from analyzing smaller, isolated organs (i.e., the heart or eyes), as our results are inherently driven by the tissues that make up the largest proportion of the postlarvae.

Similarities in the enrichment analyses of brine shrimp-reared postlarvae as compared to both wild-caught postlarvae and zooplankton-reared postarvae suggest that diet is one of the mechanisms by which brine shrimp-reared postlarvae are less robust than their wild counterparts. Transcripts related to carbohydrate metabolic process and metabolic pathways for tyrosine, serine and threonine were largely under-expressed in brine shrimp-reared postlarvae in comparison to both zooplankton-reared and wild-caught postlarvae. We interpret the under-expression of these metabolic processes in brine shrimp-reared postlarvae to mean that greater energy stores may be derived from a natural diet and/or that postlarvae may need to exert more energy to capture natural prey items, as copepods and other larval crustaceans are more difficult to catch and consume than brine shrimp [40]. This interpretation complements Castro & Cobb's 2005 study demonstrating disparate swimming speeds in wild and lab-reared postlarvae (18 cm/s in wild postlarvae compared to 10 cm/s in lab-reared postlarvae).

Further, tyrosine, serine, and threonine all have important biological functions that may be negatively impacted by the traditional lab-reared diet of brine shrimp. Tyrosine and serine are precursors for other amino acids and are also involved in neurotransmission; specifically, tyrosine is a precursor for neurotransmitters L-Dopa, dopamine, epinephrine, and norepinephrine. As such, deficiencies in serine and tyrosine are linked to impaired function of the nervous system [41]. Threonine is involved in energy production, protein synthesis, and maintaining healthy gut microbiota [42]. In rats, deficiencies in threonine compromise lipid metabolism and force a reduction in energy expenditure [43]. Again, this serves to further explain reduced energy expenditure in brine-shrimp-reared postlarvae.

Transcripts related to membrane channel activity and chitin binding were largely under-expressed in zooplankton-reared postlarvae compared to brine shrimp-reared postlarvae. Dampened membrane channel activity in zooplankton-fed postlarvae may be the result of a more varied and cholesterol-rich diet than that of postlarvae fed freshly hatched *Artemia spp*., as crustaceans are unable to synthesize their own cholesterol *de novo* and thus derive all of their cholesterol from diet [44]. A diet of zooplankton has been found to have twice the sterol content compared to a diet of *Artemia spp*. [14]. It has been documented that increased membrane cholesterol can suppress channel activity, and in particular ligand-gated channels [45], which comprise two of the top five significantly enriched GO terms in this comparison.

Transcripts related to endopeptidase activity and neuroactive ligand-receptor interaction were over-expressed in zooplankton-fed postlarvae relative to wild-caught postlarvae. Transcripts related to cell communication and signaling were under-expressed in both lab-reared groups compared to the wild-caught group, which suggests that reduced cell communication may result from factors inherent to the lab outside of diet, such as reduced stimulation or constant temperature. In crustaceans, physiological functions like hormone secretion are dictated by circadian rhythms [46] which can be disrupted by low light levels in the incubator environment.

Exposure to cold (8°C) yielded very different transcriptional responses in postlarvae from each of the rearing conditions. In brine shrimp-reared postlarvae, transcripts related to G-protein coupled receptor activity, shell-building processes, and DNA replication were largely under-expressed after exposure to 8°C, while transcripts related to the metabolism of purines,

pyrimidines, and general nucleotides were largely over-expressed. This pattern of slowed DNA replication and increased metabolism of nucleotides has been documented in other species exposed to slow growth conditions, such as colder temperatures, and may allow for cellular reorganization [47].

Transcripts related to shell-building processes were largely over-expressed in zooplankton-reared postlarvae exposed to 8˚C. This result, when compared to the trend seen in brine shrimp-reared postlarvae, may indicate that a diet of brine shrimp predisposes lobster postlarvae to be susceptible to more energetic tradeoffs in a cold-stressed state. Under-expression of transcripts related to carbohydrate metabolic processes and pathways for metabolism of serine, threonine, cysteine, and methionine was also observed in zooplankton-reared postlarvae, again illustrating reduced metabolic demands at 8˚C, in concert with the under-expression of pathways for DNA replication and glycolysis / gluconeogenesis.

Thermally induced transcriptional changes were minimal in wild postlarvae compared with either lab-reared group, with only one GO term and one KEGG pathway being significantly enriched in response to cold exposure. We interpret this decreased cellular response in wild postlarvae compared to those reared in a laboratory setting as further evidence for decreased fitness of lab-reared organisms, serving as a complement to previously described disparities in swimming ability [13] and proper nutrition [14]. Transcripts related to carbohydrate biosynthetic processes were largely over-expressed in response to chronic cold exposure in wild-caught postlarvae. The FoxO signaling pathway was the only KEGG pathway that was significantly enriched in this comparison (for sequences see S3 File). The FoxO signaling pathway has previously been identified in zebrafish as a key survival pathway for enhancing cold stress, and thus genes within it as molecular markers for thermal tolerance [48]. FoxO has also been noted for its importance to growth and stress resistance in arthropods (e.g. the brown planthopper *Nilaparvata lugens* [49]). We found a different gene in this pathway, phosphoenolpyruvate carboxykinase, that might be relevant as a genetic marker for cold tolerance in lobster. While markers for cold tolerance are often targeted in plant species for use in selective breeding practices [50, 51], they may also be valuable tools for studying crustaceans, whether assessing the quality of larvae coming from different mothers or the quality of larvae coming from a lab. Phosphoenolpyruvate carboxykinase serves in the maintenance of glucose homeostasis and converts within the gluconeogenesis pathway oxalacetate to phosphoenolpyruvate and carbon dioxide [52]. In many vertebrate ectotherms, glucose is a known cryoprotectant, preventing the denaturation of important enzymes like lactate dehydrogenase [53–55]. In crustaceans, the accumulation of other sugars and free amino acids has been linked to increased cold tolerance [56]. Thus, enhanced gluconeogenesis would increase glucose levels in the tissues and/or hemolymph and aid in cold tolerance. This gene's potential role in the cold tolerance of wild-caught lobster postlarvae is therefore fitting, though more research is needed to fully understand it.

Exposure to warm conditions (26˚C) yielded very different transcriptional responses in the two rearing conditions studied, with brine shrimp-reared postlarvae exhibiting signs of cellular stress at 26˚C and wild-caught postlarvae exhibiting no significantly enriched GO terms or pathways at all. In brine shrimp-reared postlarvae transcripts related to shell-building processes and endoplasmic reticulum were largely under-expressed after chronic exposure to 26˚C. This is similar to their response at 8˚C, and it may be that shell-building processes are de-prioritized in the face of other energetic demands at sub-optimal temperatures. In the wild, reduced shell quality could negatively impact predator avoidance as well as settlement, as shell quality is necessarily altered in stage IV lobsters for greater negative buoyancy and ease of settlement [57, 58]. Additionally, pathways for carbon metabolism, glycolysis/gluconeogenesis, starch and sucrose metabolism, and amino sugar and nucleotide sugar metabolism were all

downregulated in response to heat exposure, while the pathway for glycosaminoglycan degradation was upregulated. This pattern of gene expression in which energy is diverted from non-essential pathways toward life-saving processes is contextualized in the framework of the Oxygen- and Capacity-Limited Thermal Tolerance hypothesis (OCLTT, [59, 60]). Within the OCLTT framework critical temperatures ($Tc_{min}$ and $Tc_{max}$) are defined through the onset of internal anaerobiosis and the accumulation of anaerobic endproducts in the presence of sufficient oxygen in the environment, due to a failing circulatory or ventilatory system. Prior to reaching $Tc_{min}$ or $Tc_{max}$ a pejus temperature threshold (Tp) is characterized by a decline in fitness, growth rate, and scope for activity [61]. In the pejus range between Tp and Tc the animal's metabolism accelerates catabolic pathways producing ATP, and decelerates anabolic pathways, consuming ATP [62]. Our data of downregulating carbon metabolism, glycolysis/gluconeogenesis, starch and sucrose metabolism, and amino sugar and nucleotide sugar metabolism during heat exposure, while upregulating the pathway for glycosaminoglycan degradation indicates that 26°C lies in the pejus or sub-optimal temperature range for brine shrimp-reared postlarvae. The analysis of wild-caught postlarvae exposed to 26°C, however, did not yield any significantly enriched GO terms or KEGG pathways, indicating that they are comparatively less stressed at this temperature. Future studies should take the discrepancy between lab-reared and wild-caught larvae into account when assessing thermal tolerance or impacts of end of century temperature projections on marine larvae.

## Conclusion

Through our analysis of gene expression in postlarvae reared in different conditions, we characterized the molecular underpinnings of observed differences between traditionally lab-reared (fed brine shrimp, maintained at 18°C) and wild-caught lobster postlarvae, as well as those reared on a natural diet of zooplankton. In addition to establishing diet-driven differences in energy and amino acid metabolism, we also identified patterns in differential expression that seem to be attributable to other characteristics of a lab setting, including reduced cell communication. These findings highlight that postlarvae reared under traditional laboratory conditions may not necessarily be representative of *in situ* postlarvae; they also suggest that changing diet alone is not sufficient to eliminate cellular artefacts of lab-rearing. The gene expression profiles of postlarvae were decreasingly impacted by both cold (8°C) and warm (26°C) temperature challenges the closer their rearing conditions were to what is experienced in the wild. This finding contextualizes the results from existing research on postlarval lobster thermal tolerance as valuable but highly conservative estimations of the thermal tolerance of *in situ* organisms. We identified two gene sequences for phosphoenolpyruvate carboxykinase for further investigation as novel markers for cold tolerance in American lobster, based on their varied upregulation in our decreasingly thermally tolerant rearing groups. Further, variability in gene expression was not meaningfully greater in wild postlarvae when compared to lab-reared postlarvae, underscoring the utility of wild larvae for research purposes. In fact, future studies can be strengthened by analyzing wild early-stage organisms to better understand thermal thresholds relevant to *in situ* populations, which is of the utmost importance as coastal communities continue to gather information to better manage economically important species in the face of climate change.

## Supporting information

**S1 File. Full list of genes commonly expressed in response to cold exposure, their expression values, and results from statistical comparisons among groups.**
(XLSX)

**S2 File. Full list of genes commonly expressed in response to heat exposure, their expression values, and results from statistical comparisons among groups.**
(XLSX)

**S3 File. DNA sequences of two transcripts of PEPCK.**
(DOCX)

**S4 File. Phosphoenolpyruvate carboxykinase expression values and results from statistical comparisons among treatment groups.**
(XLSX)

## Acknowledgments

The authors thank the two anonymous reviewers for feedback which greatly improved our manuscript, Carrie Byron for her valuable structural edits to this manuscript, the DMR larval survey research and support staff for help with obtaining wild stage IV larvae, Kathleen Reardon for providing the eggers, and Julie Karlsson, Riley Fitz, Aelia Russell, and Caroline Benfer for help with rearing the larvae for the experiments.

## Author Contributions

**Conceptualization:** Aubrey Jane, Douglas B. Rasher, Markus Frederich.

**Data curation:** Aubrey Jane, Markus Frederich.

**Formal analysis:** Aubrey Jane.

**Funding acquisition:** Douglas B. Rasher, Jesica Waller, Eric Annis, Markus Frederich.

**Investigation:** Aubrey Jane.

**Methodology:** Aubrey Jane, Jesica Waller.

**Project administration:** Douglas B. Rasher, Eric Annis, Markus Frederich.

**Resources:** Jesica Waller, Eric Annis, Markus Frederich.

**Supervision:** Markus Frederich.

**Visualization:** Aubrey Jane, Markus Frederich.

**Writing – original draft:** Aubrey Jane.

**Writing – review & editing:** Aubrey Jane, Douglas B. Rasher, Jesica Waller, Eric Annis, Markus Frederich.

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
