## [Decision Letter · Decision Letter 0]

19 Apr 2024

PONE-D-24-09495Rearing condition influences gene expression in postlarval American lobster (Homarus americanus)PLOS ONE

Dear Dr. Frederich,

Thank you for submitting your manuscript to PLOS ONE. After careful consideration, we feel that it has merit but does not fully meet PLOS ONE’s publication criteria as it currently stands. Therefore, we invite you to submit a revised version of the manuscript that addresses the points raised during the review process.

We look forward to receiving your revised manuscript.

Kind regards,

Vitor Hugo Rodrigues Paiva, Ph.D.

Academic Editor

PLOS ONE

Journal Requirements:

"This work was funded by NSF grants OCE-1947639, OCE-1948108, and OCE-1948146. The authors thank the DMR larval survey research and support staff Kathleen Reardon, Robert Russell, and Carl Wilson for help with obtaining wild stage IV larvae, Kathleen Reardon for providing the eggers, and Julie Karlsson, Riley Fitz, Aelia Russell, and Caroline Benfer for help with rearing the larvae for the experiments."

Please note that funding information should not appear in the Acknowledgments section or other areas of your manuscript. We will only publish funding information present in the Funding Statement section of the online submission form. Please remove any funding-related text from the manuscript. 

4. Please note that your Data Availability Statement is currently missing the DOI/accession number of each dataset or a direct link to access each database. If your manuscript is accepted for publication, you will be asked to provide these details on a very short timeline. We therefore suggest that you provide this information now, though we will not hold up the peer review process if you are unable.

Reviewers' comments:

Reviewer's Responses to Questions

**Comments to the Author**

1. Is the manuscript technically sound, and do the data support the conclusions?

Reviewer #1: Yes

Reviewer #2: Yes

2. Has the statistical analysis been performed appropriately and rigorously? 

Reviewer #1: Yes

Reviewer #2: Yes

3. Have the authors made all data underlying the findings in their manuscript fully available?

Reviewer #1: Yes

Reviewer #2: Yes

4. Is the manuscript presented in an intelligible fashion and written in standard English?

Reviewer #1: Yes

Reviewer #2: Yes

5. Review Comments to the Author

Reviewer #1: Overall, this was a very well-written paper investigating an important facet of lobster research. I think it was designed well, the analyses appeared sound, and I anticipate it will be highly cited. My only substantive criticism is that I feel the authors should temper the language related to the value of lab-based studies. The value of this study is highlighting that lab-based studies are likely conservative, identifies some of the physiological differences that could exist (transcript differences may not reflect changes in physiological activity), and that feeding artemia likely exacerbates the artifacts and limitations of lab-based studies. While the statement is made that previous research using lab-based studies is valuable but conservative, statements indicating that future research should focus on sourcing animals from the wild may not be realistic for many questions related to thermal tolerance. There are valuable research questions that can only be addressed with lab-based studies, and there’s value to both approaches (depending on the question). It would be an overreach to suggest the take-home message should be that research on the thermal tolerance of larval lobsters need to move toward only field-based work.

Specific comments:

Line 103: Approximately how long post-isolation were larvae fed? Anecdotally, the timing of first feeding may drive long-term outcomes. It would be helpful here to understand the longest period newly hatched larvae went without feed.

Line 119: A description of how the wild lobsters were identified as stage IV would be useful here.

Line 127/129: This description is a little confusing here. It sounds like you’re saying that postlarvae fed the zooplankton diet are able to survive at 8C but the brine-shrimp postlarvae didn’t.

Line 277: I appreciate that the possible discrepancy in molting cycle was mentioned

Reviewer #2: This is a review of an article written by Jane et al entitled: Rearing conditions influence gene expression in postlarval American lobster (Homarus americanus). The grammar and writing are excellent, and it is obvious that much care was taken to make it easy to read. It discusses the issue of using lab-reared larval lobster for scientific experiments and how it could differ from the wild and other lab-reared larval studies based on the larval diets. It then examines the impact of temperature stress on lab-reared vs wild-caught larvae. I think the manuscript is incredibly strong at making the point that differences in diet and rearing conditions will impact the findings. I am not so enthusiastic on the discuss that the different impacts that temperature stress could have on the differentially reared larvae. The authors could reenforce their point that larval rearing conditions are critical and an important bias in larval studies. They could also strengthen their findings on temperature stress by focusing on the differentially expressed transcripts in common between the different rearing conditions. The commonality points to an incredible strength in the findings: even under different rearing conditions, these 34 genes were differentially expressed. WOW- tell me more about these obviously important genes/transcripts. I was genuinely excited to find out what they were.

This is good work by good scientists!

My comments are as follows:

Abstract

Highlight more important findings like the number of differentially expressed transcripts. I don’t think you should mention FoxO, but should highlight PEPCK if you think it is important. What is/are the major point(s) that you want to get across? Many scientists read only the title and then some read the abstract before they decide on reading the entire document. Hit the reader with the important findings and make the case why your paper is worth reading.

169 What biological value do you assign to over vs under expression? I personally think that noting the number of differentially expressed transcripts is what is important, but many manuscripts still note number of up and down DEG. What if you compared it the other way- then they would be reversed. This is not a correction- just something to think about. I don’t like including number of up and down in my papers; but you are your own scientist and I support your decision to do as you see best. Life would be boring if we always agreed.

173 Figure 1. It is easier to interpret the difference between volcano plots when they all have the same X and Y axis ranges. I would strongly recommend making this change- the readers will thank you (in their minds).

173 (comments for discussion) Do you think that the differential expression could be due to the size difference between the wild-caught vs brine-reared vs zooplankton-reared larvae? Do all organs and tissues maintain the same relative volume to each other in small vs larger stage IV larvae? UPDATE: Never mind- you handled this very well on line 277- well done.

179 With N=5, data variability (gene expression) can result in less ability to determine statistical differences. Do you think that the biological variability in your wild-caught larvae is greater than your lab-reared larvae and could influence the number of DEG that you find during these temperature stresses? They could be more tolerant, but I think they are more likely more variable. A heatmap could help tell you the answer if you examine the count data from the differentially expressed lab-reared animals across all animals. Do the wild animals group together or are they more variable and some group with the other rearing conditions as well?

185-259 (for discussion- GO terms) What do all of these GO terms mean in a biological sense and in terms of your experiment? Ie (3) chitin binding- is that an immune response to fungal infections or is it metabolism to digest food or is it related to changes in the lobster shell? These terms are easy to report, often reported and almost always extremely difficult to make sense of. Please explain what value you think these have to your reader. Don’t get me started on “extracellular region”. Lobsters are wonderful animals, but we just don’t know as much about them as mice or humans; hence GO should terms interpreted carefully/sceptically. Your results section is from 163-259. 96 lines. Your GO and KEGG terms are from (259-185)/96 = 0.771. I assume the vast majority of the discussion will be discussion the relevance of these GO and KEGG terms and pathways? UPDATE- I see that you have discussed them. It is hard to make a clear case.

240 Fox-O or FoxO

240 Which transcripts were under-expressed? All of the differentially expressed transcripts are interesting.

263 Why “ambient”? I saw this in figure 2 and 3 as well. I apologise if I missed the explanation for this in the materials and methods section. Do you know the wild-caught water temperature or the temperature that you held them in the lab?

277 Are you referring to the “chitin-binding” genes when you say “shell-building”? See comment above.

304 So much of the previous few paragraphs were related to metabolism and metabolic difference related to food intake or energy demands. I think you should relate this to the hepatopancreas and perhaps how much of the stage IV larvae volume this occupies (as you homogenized the whole larvae).

305 It is so hard to comment on membrane channel activity when you are looking at a whole organism. Do you think it could be specific to certain organs/tissues or everything equally? What about chitin binding as per the earlier comment? You note it here but don’t discuss it. You note that increased cholesterol can supress channel activity, but do you have evidence or reference to the fact that zooplankton is/are cholesterol rich?

313 This paragraph is too speculative. Are neuroactive receptors on all cells? Would the neuronal tissue in a whole stage IV larvae really appear as differentially expressed? Perhaps the counts for these genes can help you determine if this is the case.

322 What are the “shell-building” process transcripts? It would be great if you focused solely on the differentially expressed transcripts in common to all three rear states. Commonalities strongly suggest that they are related to the temperature stress and not the rearing conditions. Are you interested in how each of the rearing conditions impact temperature stress or how temperature stress impacts stage IV larvae? I think there is more strength here in looking at the temperature stress unrelated to rearing conditions (ie DEG in common). Figure 2. This comment also applies to the increased temperature experiment.

327 Pesky “shell-binding processes” again.

358 Could it also be that when the temperature is lower the animals feed less and therefore need to get energy from glycogen and convert it to glucose? Did you notice a difference in feeding behaviour? You also have to be careful when suggesting an enzyme in the glucose metabolism pathway is a genetic marker for thermal tolerance. There are so many physiological processes that can change metabolism and impact energy storage vs energy generation.

340 It is also tricky to say that the FoxO pathway is so important when 2 transcripts are up (both same gene or are isoforms?) and 2 are down. There are MANY enzymes in this pathway that impact many different physiological processes. 4/233 is pretty small. I like your focus on PEPCK, but I don’t think there is evidence that it affects FoxO expression and signalling.

Discussion

I think you should make a statement about using the entire organisms and how this could confound some of the findings; as some large tissues could be driving the majority of the findings. Big tissues = big counts.

6. PLOS authors have the option to publish the peer review history of their article (what does this mean?). If published, this will include your full peer review and any attached files.

Reviewer #1: No

Reviewer #2: No

---

## [Author Response · Author response to Decision Letter 0]

16 May 2024

We have visited both links and the manuscript has been formatted accordingly.

We have reviewed the methods section and confirmed that all necessary permits were obtained and disclosed, see State of Maine Department of Marine Resources Special License #2022-19-04 mentioned around line 104. We did clarify that it was the state department of marine resources that supplied us with the lobsters (thus we did not need a fishing permit).

3. Please note that funding information should not appear in the Acknowledgments section or other areas of your manuscript. We will only publish funding information present in the Funding Statement section of the online submission form. Please remove any funding-related text from the manuscript. 

Done. The new acknowledgements section is as follows: 

"The authors thank the two anonymous reviewers for feedback which greatly improved our manuscript, Carrie Byron for her valuable structural edits to this manuscript, the DMR larval survey research and support staff for help with obtaining wild stage IV larvae, Kathleen Reardon for providing the eggers, and Julie Karlsson, Riley Fitz, Aelia Russell, and Caroline Benfer for help with rearing the larvae for the experiments."

4. Please note that your Data Availability Statement is currently missing the DOI/accession number of each dataset or a direct link to access each database. If your manuscript is accepted for publication, you will be asked to provide these details on a very short timeline. We therefore suggest that you provide this information now, though we will not hold up the peer review process if you are unable.

The accession number (SRA, PRJNA1087720) is now included in the manuscript around line 155.

Done. Thank you!

5. Review Comments to the Author

Reviewer #1: Overall, this was a very well-written paper investigating an important facet of lobster research. I think it was designed well, the analyses appeared sound, and I anticipate it will be highly cited. 

Thank you for this positive assessment of the manuscript.

My only substantive criticism is that I feel the authors should temper the language related to the value of lab-based studies. The value of this study is highlighting that lab-based studies are likely conservative, identifies some of the physiological differences that could exist (transcript differences may not reflect changes in physiological activity), and that feeding artemia likely exacerbates the artifacts and limitations of lab-based studies. While the statement is made that previous research using lab-based studies is valuable but conservative, statements indicating that future research should focus on sourcing animals from the wild may not be realistic for many questions related to thermal tolerance. There are valuable research questions that can only be addressed with lab-based studies, and there’s value to both approaches (depending on the question). It would be an overreach to suggest the take-home message should be that research on the thermal tolerance of larval lobsters need to move toward only field-based work.

We agree with the reviewer and have edited our conclusion to reflect that the use of wild larvae can augment studies and provide more context for in situ dynamics as a complement to lab-based studies.

Specific comments:

Line 103: Approximately how long post-isolation were larvae fed? Anecdotally, the timing of first feeding may drive long-term outcomes. It would be helpful here to understand the longest period newly hatched larvae went without feed.

We would be very interested in learning more about this anecdotal observation, as it clearly would have significant impact on many lab studies.

Most larvae were fed within one hour of hatch (i.e. we watched a major hatch off occur and immediately removed the larvae to a bucket with feed, then isolated them). Hatches often occurred in the mornings, and larvae from these hatches were preferentially selected for rearing. Extra larvae were released back to the ocean. The absolute longest a larva would’ve gone between hatch and feed was 10 hours if they were to have hatched overnight and been isolated with larvae from the larger hatch off events. We have added this information to the manuscript.

Line 119: A description of how the wild lobsters were identified as stage IV would be useful here.

We have now specified that morphological characteristics were used to distinguish stage IV lobsters. Thank you.

Line 127/129: This description is a little confusing here. It sounds like you’re saying that postlarvae fed the zooplankton diet are able to survive at 8C but the brine-shrimp postlarvae didn’t.

We have reworded for clarity. Thank you.

Line 277: I appreciate that the possible discrepancy in molting cycle was mentioned

It’s something we’re very cognizant of. We appreciate your appreciation!

Reviewer #2: This is a review of an article written by Jane et al entitled: Rearing conditions influence gene expression in postlarval American lobster (Homarus americanus). The grammar and writing are excellent, and it is obvious that much care was taken to make it easy to read. It discusses the issue of using lab-reared larval lobster for scientific experiments and how it could differ from the wild and other lab-reared larval studies based on the larval diets. It then examines the impact of temperature stress on lab-reared vs wild-caught larvae. I think the manuscript is incredibly strong at making the point that differences in diet and rearing conditions will impact the findings. I am not so enthusiastic on the discuss that the different impacts that temperature stress could have on the differentially reared larvae. The authors could reenforce their point that larval rearing conditions are critical and an important bias in larval studies. They could also strengthen their findings on temperature stress by focusing on the differentially expressed transcripts in common between the different rearing conditions. The commonality points to an incredible strength in the findings: even under different rearing conditions, these 34 genes were differentially expressed. WOW- tell me more about these obviously important genes/transcripts. I was genuinely excited to find out what they were.

This is good work by good scientists!

The central point of this paper is to illustrate that rearing condition influences thermal tolerance, which we believe to be an important and interesting contribution to the field given the widespread use of only lab-reared marine larvae, especially given the stark contrast in genetic response to temperature among rearing groups. We have, however, briefly elaborated on the commonly expressed genes after exposure to either temperature extreme and now include supplementary tables of all of the commonly expressed genes for readers to reference as desired.

My comments are as follows:

Abstract

Highlight more important findings like the number of differentially expressed transcripts. I don’t think you should mention FoxO, but should highlight PEPCK if you think it is important. What is/are the major point(s) that you want to get across? Many scientists read only the title and then some read the abstract before they decide on reading the entire document. Hit the reader with the important findings and make the case why your paper is worth reading.

Thank you for this note. We have edited the abstract to put more numbers to the overall themes of our findings.

169 What biological value do you assign to over vs under expression? I personally think that noting the number of differentially expressed transcripts is what is important, but many manuscripts still note number of up and down DEG. What if you compared it the other way- then they would be reversed. This is not a correction- just something to think about. I don’t like including number of up and down in my papers; but you are your own scientist and I support your decision to do as you see best. Life would be boring if we always agreed.

We appreciate the food for thought. In providing the # of over- and under-expressed genes, we intend only to provide slightly more context than simply the # of differentially expressed genes.

173 Figure 1. It is easier to interpret the difference between volcano plots when they all have the same X and Y axis ranges. I would strongly recommend making this change- the readers will thank you (in their minds).

We appreciate the recommendation and see the value in a uniform scale across the different plots. However, we have chosen to format the axes this way to allow for the most granularity in the spread of individual transcripts. We highlight the different scales in the figure captions to avoid confusion.

173 (comments for discussion) Do you think that the differential expression could be due to the size difference between the wild-caught vs brine-reared vs zooplankton-reared larvae? Do all organs and tissues maintain the same relative volume to each other in small vs larger stage IV larvae? UPDATE: Never mind- you handled this very well on line 277- well done.

Thank you for your thorough read!

179 With N=5, data variability (gene expression) can result in less ability to determine statistical differences. Do you think that the biological variability in your wild-caught larvae is greater than your lab-reared larvae and could influence the number of DEG that you find during these temperature stresses? They could be more tolerant, but I think they are more likely more variable. A heatmap could help tell you the answer if you examine the count data from the differentially expressed lab-reared animals across all animals. Do the wild animals group together or are they more variable and some group with the other rearing conditions as well?

Thank you for pushing us on this! Most people would probably expect that, but it’s not the case. Wild postlarvae are not any more variable than the other groups analyzed. We used the commonly expressed genes from each temperature challenge to generate heat maps. The hierarchical clusters actually look exactly as we would expect them to, and we are confident that the number of DEGs is an accurate reflection of differences among rearing conditions and temperature treatments. We now include these heatmaps and a detailed description of them in the manuscript and are thrilled with this addition to the manuscript. Thank you!

185-259 (for discussion- GO terms) What do all of these GO terms mean in a biological sense and in terms of your experiment? Ie (3) chitin binding- is that an immune response to fungal infections or is it metabolism to digest food or is it related to changes in the lobster shell? These terms are easy to report, often reported and almost always extremely difficult to make sense of. Please explain what value you think these have to your reader. Don’t get me started on “extracellular region”. Lobsters are wonderful animals, but we just don’t know as much about them as mice or humans; hence GO should terms interpreted carefully/sceptically. Your results section is from 163-259. 96 lines. Your GO and KEGG terms are from (259-185)/96 = 0.771. I assume the vast majority of the discussion will be discussion the relevance of these GO and KEGG terms and pathways? UPDATE- I see that you have discussed them. It is hard to make a clear case.

We appreciate the point here that many of the GO terms (yes, especially “extracellular region”) are difficult to make sense of. We are looking forward to a time when there are lobster-specific GO terms (Oh the joy of working with non-model species). We do feel that there is merit to their use, though, and we took this approach to interpreting the differential expression analysis because it provided an unbiased way of combing through the troves of data that the ‘omics process yields. 

Rather than cherry-picking genes and processes, this approach allowed us to systematically assess the most significant differences between groups. This lead to a lot of interesting findings. In example, in the case of wild lobsters at extreme temperatures, it is incredibly telling to see the utter lack of terms that come up in the differential expression analysis from both the GO and KEGG databases. In addition to that finding being perfectly illustrative of the main point of this paper, we are confident that our interpretations of the meaningful terms identified are both scientifically sound (though not conclusive, and we do not present them that way) and useful for guiding new research questions.

240 Fox-O or FoxO

FoxO. Thank you!

240 Which transcripts were under-expressed? All of the differentially expressed transcripts are interesting.

In this pathway, two different serine/threonine protein kinases were under-expressed. We agree that all differentially expressed transcripts are interesting: however, there are thousands of them to go through within the already winnowed-down GO terms and KEGG pathways we present in this paper. The scope of this paper is to provide a generalized, bird’s-eye view of the differences among rearing conditions through the use of GO terms and KEGG pathways and identify a potential molecular marker for cold stress. This is why we focus on the two transcripts that were upregulated in response to cold stress in our wild lobsters.

263 Why “ambient”? I saw this in figure 2 and 3 as well. I apologise if I missed the explanation for this in the materials and methods section. Do you know the wild-caught water temperature or the temperature that you held them in the lab?

We chose to maintain the wild postlarvae at ambient temperature in the lab so that there would not be an added artefact of stress from introducing them to an artificially warmer and constant temperature. Or in other words, we wanted all of the control conditions to be what the lobsters were reared at. This was done by housing wild postlarvae’s individual jars in a flow through seawater bath, and the temperatures fluctuated from 11-19dC throughout the summer as noted around line 122, as it would’ve if they were still in the ocean.

277 Are you referring to the “chitin-binding” genes when you say “shell-building”? See comment above.

Yes, in conjunction with structural constituents of the cuticle. We now specify this for clarity.

304 So much of the previous few paragraphs were related to metabolism and metabolic difference related to food intake or energy demands. I think you should relate this to the hepatopancreas and perhaps how much of the stage IV larvae volume this occupies (as you homogenized the whole larvae).

We have included a sentence highlighting the limitations of analyzing whole-body samples. However, we feel that attributing any specific results to expression within particular organs is too speculative and beyond the scope of this study. As it stands right now, we have no reason to believe that at the same stage, different rearing conditions yield different proportions of organs within the body cavity. Our team is working on a separate effort, developi

---

## [Decision Letter · Decision Letter 1]

9 Jun 2024

PONE-D-24-09495R1Rearing condition influences gene expression in postlarval American lobster (Homarus americanus)PLOS ONE

Dear Dr. Frederich,

Thank you for submitting your manuscript to PLOS ONE. After careful consideration, we feel that it has merit but does not fully meet PLOS ONE’s publication criteria as it currently stands. Therefore, we invite you to submit a revised version of the manuscript that addresses the points raised during the review process.

We look forward to receiving your revised manuscript.

Kind regards,

Vitor Hugo Rodrigues Paiva, Ph.D.

Academic Editor

PLOS ONE

Journal Requirements:

Reviewers' comments:

Reviewer's Responses to Questions

**Comments to the Author**

1. If the authors have adequately addressed your comments raised in a previous round of review and you feel that this manuscript is now acceptable for publication, you may indicate that here to bypass the “Comments to the Author” section, enter your conflict of interest statement in the “Confidential to Editor” section, and submit your "Accept" recommendation.

Reviewer #1: All comments have been addressed

Reviewer #3: (No Response)

2. Is the manuscript technically sound, and do the data support the conclusions?

Reviewer #1: Yes

Reviewer #3: Yes

3. Has the statistical analysis been performed appropriately and rigorously? 

Reviewer #1: Yes

Reviewer #3: Yes

4. Have the authors made all data underlying the findings in their manuscript fully available?

Reviewer #1: Yes

Reviewer #3: Yes

5. Is the manuscript presented in an intelligible fashion and written in standard English?

Reviewer #1: Yes

Reviewer #3: Yes

6. Review Comments to the Author

Reviewer #1: (No Response)

Reviewer #3: General feedback:

This is a review of the revised manuscript, “Rearing condition influences gene expression in postlarval American lobster (Homarus americanus),” submitted by Jane et al. to PLOS ONE. This is an interesting paper that explores the differential gene expression of lab-reared vs. wild-caught postlarval lobster in the context of diet and thermal stress. This is a well-written and focused paper that contributes to our broader understanding of lobster biology while providing a strong evidence for scientists to consider how diet and rearing conditions (i.e., lab-reared vs. wild-caught postlarvae) impact experimental results. The authors do an excellent job addressing the insightful comments/feedback from the previous reviewers, which I strongly believe helped to improve the overall message and conclusions of the manuscript. The authors managed to maintain their point of view while ensuring the results were clearly explained through the inclusion of additional and/or modified figures and tables.

I do have a few minor suggestions (see below), but overall think this was an excellent revision that addressed the reviewers’ feedback while strengthening the central conclusions – well done!

Specific feedback:

Abstract:

• Lines 28-30: I agree that this should be included in the abstract, but find it a bit hard to read. Perhaps specify “wild-caught”? It also seems odd to start a sentence with a number (line 29)?

• Line 33: Since the abstract is what draws readers in, perhaps you could include a description of the FoxO signaling pathway.

Introduction:

Line 59: The way this sentence is written suggests that development time increases (i.e., is longer) with warming, but you clearly state that development time is reduced (i.e., faster) under warming at the end of the paragraph – perhaps re-word to clarify your point?

Methods:

• Line 97: Should Waller et al. (2023) be a numerical citation?

• Lines 111-112: It would be helpful to include the average sizes rather than just reporting the percent increase in size of postlarvae fed zooplankton vs. brine shrimp.

• Line 119: I think there is a typo (misplaced period).

• Line 120: Is ambient referring to temperature at the collection site?

• Lines 121-122: As stated above, it would be helpful to include average sizes of postlarvae in different treatments rather than just the percent increase in size of wild-caught individuals.

• Line 126: I think you took the explanation out, but why did you only expose zooplankton-fed postlarvae to cold stress?

• Lines 129-130: You might consider adding a sentence explaining how different the temperature treatments were from natural conditions. I’d also explain what “immediate” means - did you just change the heater to 26°C from 18 (or lower), or was there an increase/decrease over a day? I imagine it could be really stressful to go through immediate shifts, and that this might influence your results?

• Line 139: Should this be a numerical citation?

Results:

• Line 182: I would spell out the number to start the sentence.

• Line 213: I’m not sure I understand why you called out heat shock protein 90 here when it’s not mentioned in the discussion?

• Lines 234, 255, and 266: It would be helpful to clarify that this was at ambient temperature in the subheadings.

Discussion:

• Line 383: This seems incomplete – maybe change to “wild-caught” or “wild postlarvae”.

• Line 414: Should “Zhang et al. 2023” be a numerical citation?

• Line 446: I think there may be a typo (misplaced underscore between “data of”).

7. PLOS authors have the option to publish the peer review history of their article (what does this mean?). If published, this will include your full peer review and any attached files.

Reviewer #1: No

Reviewer #3: No

---

## [Author Response · Author response to Decision Letter 1]

28 Jun 2024

We thank the reviewers for the comments and suggestions and outline below how we addressed them:

6. Review Comments to the Author

Reviewer #3: General feedback:

This is a review of the revised manuscript, “Rearing condition influences gene expression in postlarval American lobster (Homarus americanus),” submitted by Jane et al. to PLOS ONE. This is an interesting paper that explores the differential gene expression of lab-reared vs. wild-caught postlarval lobster in the context of diet and thermal stress. This is a well-written and focused paper that contributes to our broader understanding of lobster biology while providing a strong evidence for scientists to consider how diet and rearing conditions (i.e., lab-reared vs. wild-caught postlarvae) impact experimental results. The authors do an excellent job addressing the insightful comments/feedback from the previous reviewers, which I strongly believe helped to improve the overall message and conclusions of the manuscript. The authors managed to maintain their point of view while ensuring the results were clearly explained through the inclusion of additional and/or modified figures and tables.

Thank you for this positive assessment of our revised manuscript. The reviewers’ comments were most helpful in strengthening it.

I do have a few minor suggestions (see below), but overall think this was an excellent revision that addressed the reviewers’ feedback while strengthening the central conclusions – well done!

Specific feedback:

Abstract:

• Lines 28-30: I agree that this should be included in the abstract, but find it a bit hard to read. Perhaps specify “wild-caught”? It also seems odd to start a sentence with a number (line 29)?

We adjusted the wording to make it easier readable

• Line 33: Since the abstract is what draws readers in, perhaps you could include a description of the FoxO signaling pathway.

We included a very brief description of the FoxO pathway in the abstract

Introduction:

Line 59: The way this sentence is written suggests that development time increases (i.e., is longer) with warming, but you clearly state that development time is reduced (i.e., faster) under warming at the end of the paragraph – perhaps re-word to clarify your point?

The word “decreased” was missing for the development time. Thanks for catching this

Methods:

• Line 97: Should Waller et al. (2023) be a numerical citation?

We removed this citation as it is not necessary

• Lines 111-112: It would be helpful to include the average sizes rather than just reporting the percent increase in size of postlarvae fed zooplankton vs. brine shrimp.

We now include the average weight of both groups of larvae

• Line 119: I think there is a typo (misplaced period).

fixed

• Line 120: Is ambient referring to temperature at the collection site?

Temperature of the flow-through tank fed by the Damariscotta River estuary; we clarify this now in the manuscript

• Lines 121-122: As stated above, it would be helpful to include average sizes of postlarvae in different treatments rather than just the percent increase in size of wild-caught individuals.

We now include the average weight of both groups of larvae

• Line 126: I think you took the explanation out, but why did you only expose zooplankton-fed postlarvae to cold stress?

This was done simply for logistical reasons (limited time and funding for additional experiments). At the time when we made the decision, it seemed a good choice. After analyzing the data, we certainly wished that we had done these experiments as well, but now it is unfortunately too late. We refrained from explaining this (difficult) choice in the manuscript.

• Lines 129-130: You might consider adding a sentence explaining how different the temperature treatments were from natural conditions. I’d also explain what “immediate” means - did you just change the heater to 26°C from 18 (or lower), or was there an increase/decrease over a day? I imagine it could be really stressful to go through immediate shifts, and that this might influence your results?

We added the respective details

• Line 139: Should this be a numerical citation?

Fixed

Results:

• Line 182: I would spell out the number to start the sentence.

We reworded the sentence so that it does not start with a number

• Line 213: I’m not sure I understand why you called out heat shock protein 90 here when it’s not mentioned in the discussion?

We removed the mentioning of HSP70 here

• Lines 234, 255, and 266: It would be helpful to clarify that this was at ambient temperature in the subheadings.

Adding this information into the subheadings would make them very long. Therefore, we added the description of this being larvae reared at ambient temperature into the first sentences of the respective paragraphs, following the subheadings.

Discussion:

• Line 383: This seems incomplete – maybe change to “wild-caught” or “wild postlarvae”.

Fixed

• Line 414: Should “Zhang et al. 2023” be a numerical citation?

Fixed

• Line 446: I think there may be a typo (misplaced underscore between “data of”).

Fixed

---

## [Editor Report · Decision Letter 2]

2 Jul 2024

Rearing condition influences gene expression in postlarval American lobster (Homarus americanus)

PONE-D-24-09495R2

Dear Dr. Frederich,

We’re pleased to inform you that your manuscript has been judged scientifically suitable for publication and will be formally accepted for publication once it meets all outstanding technical requirements.

Kind regards,

Vitor Hugo Rodrigues Paiva, Ph.D.

Academic Editor

PLOS ONE
---

## [Editor Report · Acceptance letter]

10 Jul 2024

PONE-D-24-09495R2 

PLOS ONE

Dear Dr. Frederich, 

I'm pleased to inform you that your manuscript has been deemed suitable for publication in PLOS ONE. Congratulations! Your manuscript is now being handed over to our production team.

Kind regards, 

on behalf of

Dr. Vitor Hugo Rodrigues Paiva 

Academic Editor

PLOS ONE